# Extracellular Matrix Deposition and Remodeling after Corneal Alkali Burn in Mice

**DOI:** 10.3390/ijms22115708

**Published:** 2021-05-27

**Authors:** Kazadi N. Mutoji, Mingxia Sun, Garrett Elliott, Isabel Y. Moreno, Clare Hughes, Tarsis F. Gesteira, Vivien J. Coulson-Thomas

**Affiliations:** 1College of Optometry, University of Houston, Houston, TX 77204, USA; kmutoji@central.uh.edu (K.N.M.); msun9@central.uh.edu (M.S.); gelliott.2021@alumni.opt.uh.edu (G.E.); iymoreno@cougarnet.uh.edu (I.Y.M.); tarsis.ferreira@gmail.com (T.F.G.); 2School of Biosciences, Cardiff University, Cardiff CF10 3AT, UK; HughesCE1@cardiff.ac.uk; 3Optimvia, Batavia, OH 45103, USA

**Keywords:** glycosaminoglycans, proteoglycans, corneal stroma, scar formation, corneal wounding

## Abstract

Corneal transparency relies on the precise arrangement and orientation of collagen fibrils, made of mostly Type I and V collagen fibrils and proteoglycans (PGs). PGs are essential for correct collagen fibrillogenesis and maintaining corneal homeostasis. We investigated the spatial and temporal distribution of glycosaminoglycans (GAGs) and PGs after a chemical injury. The chemical composition of chondroitin sulfate (CS)/dermatan sulfate (DS) and heparan sulfate (HS) were characterized in mouse corneas 5 and 14 days after alkali burn (AB), and compared to uninjured corneas. The expression profile and corneal distribution of CS/DSPGs and keratan sulfate (KS) PGs were also analyzed. We found a significant overall increase in CS after AB, with an increase in sulfated forms of CS and a decrease in lesser sulfated forms of CS. Expression of the CSPGs biglycan and versican was increased after AB, while decorin expression was decreased. We also found an increase in KS expression 14 days after AB, with an increase in lumican and mimecan expression, and a decrease in keratocan expression. No significant changes in HS composition were noted after AB. Taken together, our study reveals significant changes in the composition of the extracellular matrix following a corneal chemical injury.

## 1. Introduction

The cornea is a transparent tissue that forms the outermost part of the eye and is responsible for the initial refraction of light from the external environment into the eye, and ultimately onto the retina, resulting in the manifestation of an image [1,2,3]. Transparency, defined as the capacity that the cornea possesses to transmit light without significant scattering, is a unique physical property of the cornea that is essential for vision [4,5,6]. The cornea is formed of a highly organized dense connective tissue, the stroma, flanked on the outer side by stratified squamous epithelia, the epithelium, and on the inner side by simple epithelia, the endothelium [7,8]. The structural makeup of the cornea directly correlates with its function. The corneal stroma comprises approximately 90 percent of the total corneal thickness and provides the majority of its structural framework [6,8,9,10,11,12]. The stroma is composed of a highly organized network of largely Type I and V collagen fibrils with an extracellular matrix (ECM) consisting of four main structural components, water, salts, glycoproteins, and proteoglycans (PGs) [3,13]. The types of specialized PGs that are primarily present in the corneal stroma include lumican, keratocan, mimecan, decorin, byglican, and versican [14,15,16,17]. Collagen fibrils in the stroma form highly organized parallel layers, namely the lamella, which ensure corneal transparency [7,18]. Small leucine rich proteoglycans (SLRPs) have been shown to regulate both collagen spacing and orthogonal organization in the cornea [7,18]. Lumican, keratocan, mimecan, decorin, and biglycan are all members of the SLRP family of PGs. SLRPs are composed of a core protein, 25–65 kDa in size characterized by tandem arrays of leucine-rich repeats flanked by cysteine-rich amino and carboxyl domains with putative covalently bound glycosaminoglycans (GAGs) [19,20,21]. GAGs are composed of repeating disaccharide units containing sulfate esters and can be divided into six main subtypes: keratan sulfate (KS), chondroitin sulfate (CS), dermatan sulfate (DS), heparan sulfate (HS), heparin (HEP), and hyaluronan (HA). An additional subtype, acharan sulfate, has been identified in the mollusk [22]. The corneal stroma is one of the mammalian tissues with the highest KS content, and is the primary tissue that contains lumican, keratocan and mimecan in the KSPG form [23,24,25,26,27,28]. The corneal stroma also contains significant levels of CS/DSPGs, decorin, biglycan, and versican [23,24,25,26,27,28]. Numerous studies over the years have established the importance of PGs and GAGs in maintaining corneal transparency, specifically, in regulating collagen fibril diameter, interfibrillar spacing, and intrafibrillar spacing [18,26]. The Chakravarti group was able to demonstrate that lumican *null* mice present corneal opacity as a result of increased fibril diameter, altered fibril structures and increased interfibrillary spacing, all due to a reduction in total KSPG levels [29,30]. A similar increase in collagen fibril diameter was also observed in a different study using mimecan *null* mice [21]. Since these early findings many other studies have further shown that KSPGs and CS/DSPGs play a key role in establishing and maintaining corneal transparency by maintaining structural integrity of collagen fibrils [18,31,32].

As with any tissue that is exposed to the external environment, the cornea is subject to various insults, which can lead to injuries. Injuries from external sources can range from abrasions and/or blunt traumas to chemical burns from alkali household cleaners such as ammonia, sodium hydroxide, plaster, and cement. Chemical burns are the most common form of corneal injuries observed in a clinical setting [33]. Unfortunately, many of these injuries lead to permanent corneal opacification due to significant stromal scarring and angiogenesis, and, as a consequence, this loss of transparency leads to impaired vision. There are few available treatment options to prevent/limit corneal scarring, which is caused by excessive ECM deposition and angiogenesis during the process of wound healing. Severe corneal scarring and angiogenesis after injury can ultimately lead to a loss of vision. Loss of corneal transparency is one of the leading causes of blindness worldwide [34]. Thus, more research is needed to understand mechanisms of limiting scarring after chemical wounds. For such, it is necessary to understand the process of ECM deposition after injury, and how it affects corneal transparency. Depending on the severity, size, and location of the corneal scar, corneal transplantation may be the only potential method of treatment to restore vision. As with any organ transplant, this requires finding and/or waiting for a donor (which are currently limited by the increased popularity of LASIK surgery) and the use of long-term immune suppressors. Thus, there is a clear unmet medical need for new treatments for preventing and/or treating corneal scarring.

Keratocytes are the most abundant cell type found in the corneal stroma, occupying 9 to 17 percent of its total volume [3,6,35]. Keratocytes are fibroblast-like cells that secrete and maintain the stromal ECM both during homeostasis and after injury [3,36]. After corneal injury, keratocytes are induced into the myofibroblast phenotype, which leads to increased proliferation and ECM deposition [37]. This cellular trans-differentiation associated with increased and altered ECM deposition forms a provisional matrix that supports the wound healing process; however, this in turn alters the normal collagen organization, leading to corneal scaring, which, when excessive, can compromise vision [36,38,39]. A key aspect of the wound healing process, and a key point of interest for therapeutic industries, is to limit ECM deposition after corneal injury and promote ECM remodeling, both during and after wound healing, to restore the original collagen organization necessary for corneal transparency [40]. GAGs and PGs are known to be present in the provisional matrix after wounding, and are key in maintaining the collagen organization that is necessary for restoring corneal transparency [41]. In order to better understand how the composition of the corneal ECM changes after injury, and, throughout the wound healing process, we carried out a thorough investigation to characterize the expression and distribution of the different GAGs and PGs in mouse corneas after a chemical injury using a standardized mouse alkali-burn (AB) model.

## 2. Results

### 2.1. Alkali Burn Leads to Corneal Scarring

Corneas were imaged under a stereomicroscope before (uninjured), immediately after (0 h), 5 days after (5 days), and 14 days after (14 days) AB. Significant corneal clouding was evident in all corneas 5 and 14 days after AB (representative images shown in Figure 1A). Corneal scarring was evaluated by in vivo confocal microscopy using a Heidelberg Retinal Tomograph—HRTII with the Rostock Cornea Module. In vivo confocal microscopy allows high magnification images to be obtained at a cellular level throughout the cornea of live mice. With this form of analysis, quantitative measurements can be obtained for corneal thickness, extracellular matrix backscatter, changes in corneal keratocyte density/morphology, and inflammatory cell infiltration. Representative images at the depth of the anterior and posterior stroma of corneas 5 and 14 days after AB are displayed in Figure 1B. Significant corneal scarring was evident throughout the corneal stroma at 5 and 14 days after AB (Figure 1B). Moreover, inflammatory cell infiltration was clearly evident throughout the corneal stroma 5 days after AB (Figure 1B, top panel). Corneal backscatter (which directly correlates with corneal haze) was quantified throughout the full corneal thickness and plotted in a graph. There was a significant increase in corneal haze immediately, 5 days and 10 days after AB (Figure 1C). Thus, this data indicates that, as previously shown, the mouse cornea is significantly inflamed at 5 days after AB, and corneal scarring is evident at both 5 days and 14 days after AB [42]. Herein we analyzed and compared the ECM composition of the mouse cornea during the process of wound healing following an AB. Corneas 5 and 14 days after AB were compared to the contralateral uninjured eye.

### 2.2. Characterization of HS/HEP in Murine Corneas after Alkali Burn

GAGs were isolated from injured and uninjured mouse corneas 5 and 14 days after AB, and HS disaccharides were generated by heparinase I and III digestion and analyzed by strong anion chromatography (SAX) using high-pressure liquid chromatography (HPLC). Overall, the analysis revealed that summed abundances of HS disaccharides were similar in injured and uninjured corneas both 5 and 14 days after injury (Table 1). When analyzing the structural composition of HS in the cornea, the most abundant disaccharide identified was, D0A0 (ΔUA-GlcNAc), followed by D0S0 (ΔUA-GlcNS), D2S0 (ΔUA2S-GlcNS), D0A6 (ΔUA-GlcNAc6S), D2S6 (ΔUA2S-GlcNS6S), and, finally, in significantly lower abundance, D0S6 (ΔUA-GlcNS6S), D2A6 (ΔUA2S-GlcNAc6S), and D2A0 (ΔUA2S-GlcNAc) (Figure 2 and Table 1). It is worthy of note that the abundance of N-acetylated disaccharides was slightly higher than that of N-sulfated disaccharides. Thus, in general, HS has an overall low sulfation pattern in the cornea with an average of one sulfate per disaccharide unit. It is also worthy of note that 3-O-sulfated disaccharides were not identified in the cornea. Overall, no significant changes in HS composition were noted at 5 or 14 days after AB when compared to the uninjured contralateral eye (Figure 2). Therefore, no significant changes in HS levels or composition were observed in the corneas after AB.

### 2.3. Characterization of CS/DS in Murine Corneas after Alkali Burn

For the analysis of CS/DS disaccharide units in injured and uninjured mouse corneas 5 and 14 days after AB, GAGs were digested with Chondroitinase ABC and, thereafter, analyzed by SAX-HPLC. Firstly, in contrast to HS, there was a significant increase in the summed abundances of CS/DS disaccharides between the injured and uninjured corneas both 5 and 14 days after injury (Table 2). Specifically, there was a ~25% increase in total CS 5 days after AB and a ~40% increase in total CS 14 days after AB (Table 2). The most abundant disaccharide identified was D2a4 (ΔUA2S-GalNAc4S), and, interestingly, it increased in abundance by almost 2-fold 14 days after alkali burn (Figure 3 and Table 2). Thus, 2,4-sulfated CS is highly expressed in the murine cornea, and is up-regulated 14 days after AB (Figure 3). Next, D0a4 (ΔUA-GalNAc4S) and D0a0 (ΔUA-GalNAc) are the most abundant disaccharides in the uninjured tissue, with similar abundances (Figure 3). The abundance of D0a4 does not change after AB, while the abundance of D0a0 significantly decreases after AB (Figure 3). D0a6 (ΔUA-GalNAc6S) is the next most abundant CS/DS disaccharide in the uninjured cornea with its abundance significantly decreasing after AB (Figure 3). Finally, the least abundant CS/DS disaccharide identified in the uninjured cornea was D0A10 (ΔUA-GalNAc4S6S), and, interestingly, this disaccharide was significantly more abundant after AB; specifically, there was a ~5-fold increase 5 days after AB, and at 14 days after AB, it went from being undetected in the contralateral uninjured cornea to being abundantly present in the injured cornea (Figure 3). Thus, in general, there is a decrease in the abundance of low sulfated CS/DS disaccharides after AB and an increase in highly sulfated disaccharides. Thus, highly sulfated CS could potentially play a role in the provisional matrix during wound healing, and, also, be part of the corneal scarring process. It is worthy of note that the disaccharides D2a0 (ΔUA2S-GalNAc), D2a6 (ΔUA2S-GalNAc6S), and D2a10 (ΔUA2S-GalNAc4S6S), were not identified in the naïve or injured murine corneas.

### 2.4. Distribution of CS Sulphated Epitopes in Murine Corneas after Alkali Burn

The distribution of CS was also analyzed in corneas 14 days after AB using specific anti-CS antibodies. For this analysis, anti-CS antibodies, clones 1B5, 3B3, and 2B6, were used following Chondroitinase ABC treatment of the tissues. Clone 3B3 recognizes a disaccharide containing a glucuronic acid adjacent to an N-acetylgalactosamine-6-sulfate at the non-reducing terminal of native CS (3B3-epitope) and a disaccharide containing an unsaturated uronic acid adjacent to a 6-sulfated N-acetylgalactosamine at the non-reducing terminal of Chondroitinase ABC digested CS stubs (3B3+ epitope) [43]. Clone 1B5 recognizes a non-sulfated disaccharide containing an unsaturated uronic acid adjacent to an N-acetylgalactosamine and, clone 2B6 recognizes a disaccharide containing an unsaturated uronic acid adjacent to a 4-sulfated N-acetylgalactosamine, both at the non-reducing terminal of Chondroitinase ABC digested CS stubs. Our data show that uninjured corneas present strong 1B5 staining in the anterior stroma, limited staining in both the posterior stroma and endothelium and no staining in the corneal epithelium (Figure 4A). Fourteen days after alkali burn, strong 1B5 staining is present throughout the stroma and endothelium, but no staining is present in the corneal epithelium (Figure 4A). When observing the limbal region of uninjured corneas, a decrease in 1B5 staining can be noted throughout the stroma in the outer peripheral cornea when compared to the central cornea. In contrast, 1B5 staining is present throughout all layers of the limbal stroma and endothelium, with stronger staining in the anterior stroma (Figure 4C). Fourteen days after AB, there is an increase in 1B5 staining in the endothelium, anterior stroma, and epithelium in the outer peripheral cornea, and an increase in 1B5 staining in the anterior stroma and epithelium in the limbal region (Figure 4C). When analyzing the 3B3 staining pattern of Chondroitinase ABC digested tissues, limited 3B3+ staining was observed in uninjured corneas which was limited to the anterior stroma (Figure 4A). Fourteen days after AB, an increase in 3B3+ staining was noted throughout all layers of the cornea, epithelium and endothelium (Figure 4A). In the limbal region of uninjured corneas, 3B3+ staining was observed in the endothelium and in patches of the anterior stroma (Figure 4C). An increase in 3B3+ staining was noted in the endothelium, anterior stroma and epithelium of the peripheral cornea and limbus 14 after AB (Figure 4C). 2B6 staining was found exclusively in the stroma of uninjured corneas, primarily in the anterior stroma (Figure 4B). 14 days after AB, an increase in 2B6 staining was observed throughout all layers of the corneal stroma and endothelium, and punctate staining could be observed in superficial layers of the epithelium (Figure 4B). In the limbus, 2B6 staining was observed throughout the stroma, endothelium and basal layer of the epithelium of uninjured corneas and was increased 14 days after AB (Figure 4D).

### 2.5. Changes in CSPG Expression 5 and 14 Days after Alkali Burn in Murine Corneas

Given the significant changes in CS expression in the corneas after AB, we also analyzed the expression profile of various CS/DSPGs. There was a significant increase in biglycan expression both 5 and 14 days after AB, while there was a decrease in decorin expression 5 days (*p* ≤ 0.05) and 14 days (did not reach significance) after AB (Figure 5D,E). Biglycan expression and localization was also analyzed in uninjured corneas and corneas 14 days after AB by immunofluorescence. In uninjured corneas, biglycan is present in the basement membrane, in the inter- and intrafibrillar space and surrounding the endothelial cells, with an increase in these locations 14 days after AB (Figure 6B). In the limbal region, biglycan had a similar distribution pattern as noted in the central cornea; it was present in the basement membrane, in the inter- and intrafibrillar space and surrounding endothelial cells (Figure 6D). There was a significant increase in biglycan within the limbal region 14 days after AB, primarily in the anterior stroma, basement membrane, and epithelium (Figure 6D).

The expression profile of the CSPG versican was also analyzed before and after AB. There was a significant increase in versican expression after AB, specifically a 6-fold increase in expression 5 days after AB and 1.5-fold increase in expression 14 days after AB (Figure 5F). Thereafter, we investigated the expression profile of the different versican isoforms after AB (Figure 5G–K). Curiously, the V0, V1, and V3 isoforms followed similar expression patterns, with increased expression 5 and 14 days after AB, although with a significantly higher increase in expression 5 days after AB, which was ~7 fold for each isoform, compared to a ~2-fold increase at 14 days after AB. For the V2 isoform, there was an increase in expression 5 days after AB, but this did not reach significance, and there were no differences in expression levels 14 days after AB. Based on the qPCR data analyzed via the 2^−ΔCt^ method, V1 was identified as the most highly expressed isoform in the cornea, followed by V0, V3, and finally V2. The expression of V1 was 5-fold higher than V0, the expression V0 was 14-fold higher than V3, and V3 was 2-fold higher than V2 in naïve corneas. Versican distribution was also analyzed in uninjured corneas and corneas 14 days after AB by immunofluorescence of tissues digested or not digested with Chondroitinase ABC. In tissues digested with Chondroitinase ABC, limited versican (stained with anti-versican AF3054) expression was noted throughout uninjured corneas (Figure 4B). An increase in versican staining was observed 14 days after AB which was present primarily in the basement membrane (Figure 4B). In the peripheral cornea and limbal region, versican staining was observed solely around isolated keratocytes (Figure 4D). A slight increase in versican staining was noted 14 days after AB as punctate staining primarily in the posterior stroma (Figure 4D). In tissues not digested with Chondroitinase ABC, limited versican was identified throughout the central cornea with clone AF3054, while versican was identified throughout the cornea with clone ab177480 (Figure 6A,B, respectively). Fourteen days after AB, there was an increase in versican expression throughout all layers of the cornea with clone AF3054 (Figure 6A). In contrast, there was solely an increase in versican expression in the corneal epithelium and surrounding a few cells within the stroma with clone ab177480 (Figure 6B). In the limbal region, versican was identified throughout all layers of the cornea of uninjured corneas, with an increase in these same regions 14 days after AB, with clone AF3054 (Figure 6C). High levels of versican were detected throughout all corneal layers in the limbal region of uninjured corneas with clone ab177480, which was mostly limited to the epithelium and to stromal cells 14 days after AB (Figure 6D). Finally, aggrecan expression was also analyzed by immunohistochemistry (Figure 6A,C). No aggrecan staining was identified in the central cornea and limbal region of uninjured corneas, whereas aggrecan was identified in the epithelium, surrounding stromal cells and in the endothelium of corneas 14 days after AB (Figure 6A,C).

### 2.6. Expression of KS and KSPS after Alkali Burn in Murine Corneas

Given the lack of enzymes for complete digestion of KS that would allow compositional analysis, we analyzed the distribution of KS throughout the corneas using immunohistochemistry with two anti-KS antibodies, clone 5D4, which recognizes a highly sulfated KS, and 1B4, which recognizes a lesser sulfated KS. In the cornea, KS was detected throughout the stromal and endothelial layers, but no KS was detected within the corneal epithelium (Figure 7A,B). Similar expression profiles were observed for both highly and lesser sulfated KS, specifically, higher expression of KS was observed in the anterior stroma when compared to the posterior stroma (Figure 7A,B). Fourteen days after AB, an increase in KS was observed throughout the cornea, including within the epithelium. Similar expression profiles were also observed for the 5D4 and 1B4 epitopes 14 days after AB (Figure 7A,B). When comparing the distribution of KS in the limbal region to the central cornea, a decrease in the highly sulfated form of KS (5D4 epitope) was observed in the anterior stroma (Figure 7C,D). In contrast, an increase in the expression of a lesser sulfated form of KS (the 1B4 epitope) was observed in the anterior stroma of the limbal region when compared to the central cornea (Figure 7C,D). Similar to what was observed with the central cornea, there was a significant increase in KS expression throughout all corneal layers in the limbal region 14 days after AB.

In order further understand the expression pattern of KS in the murine cornea, we analyzed the expression levels of different KS biosynthetic enzymes in uninjured wild-type corneas by analyzing RNA-seq data that was generated comparing the central cornea to the limbal region of naïve wild-type mice. Interestingly, 11 genes involved in KS biosynthesis were found to be differently expressed between the central and limbal region (Table 3). The genes identified encode enzymes associated with the biosynthesis of both the N-glycan (KS-I) and O-glycan (KS-II). This includes five of the seven beta-1,4-galactosyltransferases (B4galt) and two genes that catalyze the transfer of sulfate groups to KS, carbohydrate 6-sulfotransferases (Chst) 1 and 2. B4GALT family of enzymes are involved in the polymerization of the KS chain, while CHSTs catalyze the transfer of sulfate to the position 6 of either galactose or acetylglucosamine of KS [46,47,48]. We then further analyzed the expression profile of Chst1, Chst2, B4galt1, and B4galt4, in corneas 5 and 14 days after AB, compared to the uninjured contralateral control eye by real-time PCR (Figure 8). A decrease in the expression of Chst2 and B4galt 1 and 4 was observed at 5 days after AB (Figure 8B,C). Both B4galt 1 and 4 are involved in the transfer of a galactose residue to the polymerizing KS chain, indicating there is an initial decrease in the biosynthesis of KS 5 days after AB. The expression of B4galt 1 and 4 appears to rebound at 14 days, indicating that between 5 and 14 days after AB there is an increase in KS expression (Figure 8C,D). The expression of Chst1 significantly increased at both 5 and 14 days post injury, moreover, the expression of Chst2 is significantly increased at 14 days after AB, when compared to the contralateral uninjured eye (Figure 8A,B). Thus, our data indicates that, both 5 and 14 days after AB burn there is an overall increase in the sulfation of KS. Taken together, expression levels of the KS biosynthetic enzymes support our immunohistochemistry data, which shows an increase in KS at 14 days after AB. Curiously, the expression of CHST2 was higher in the contralateral control eye at five days after AB when compared to the contralateral control at 14 days after AB (Figure 8B). Previous studies have shown that in certain instances a corneal injury can trigger an immune response in the contralateral uninjured cornea [49,50,51,52,53,54,55]. Thus, the corneal injury could lead to changes in CHST2 expression in the contralateral eye over time. Importantly, we did not observe any other differences between the contralateral eye 5 and 14 days after AB in this study.

Finally, we analyzed the expression profile of different KSPGs. Curiously, we did not find any changes in the expression levels of lumican 5 days after AB, but there was a ~1.4-fold increase in lumican expression 14 days after AB that did reach significance (Figure 5A). The distribution of lumican was also analyzed by immunohistochemistry. Lumican was identified throughout all corneal layers in both uninjured and injured corneas, but at increased levels in the latter (Figure 7B). Interestingly, the increase in lumican expression after AB was not at the same magnitude as that observed for KS, thus other KSPGs would have to also be upregulated following AB or there could be an overall increase in the number of putative KS chains per lumican molecule 14 days after AB. A striking decrease in keratocan expression was observed, both 5 and 14 days after AB, specifically a 10-fold and 5-fold decrease, respectively. A subtle, however significant, increase in mimecan expression was noted both 5 and 14 days after AB when compared to the uninjured contralateral control.

## 3. Discussion

In this study, we used the well-established AB mouse model to assess the change in GAG composition during corneal wound healing 5 and 14 days after a chemical injury. This model leads to inflammatory cell infiltration and corneal scarring, and the loss of transparency is evident at both 5 and 14 days after AB [56,57,58,59,60]. Disaccharide analysis revealed there was a significant increase in CS expression after AB, and, thus, CSPGs are important constituents of the provisional matrix in the cornea. Importantly, the CS chains that are expressed following AB are more highly sulfated than those expressed in the uninjured cornea. Thus, highly sulfated CS could potentially play a role in the provisional matrix during wound healing and also be part of the corneal scarring process. Interestingly, CS/DS sulfatases are a family of enzymes that catalyze the hydrolysis of the sulfate groups on CS/DS and have been identified in many mammalian and bacterial species [61]. Further studies would have to establish whether targeting sulfated CS with extracellular sulfatases could be used as a pharmaceutical target for limiting corneal scarring after injury.

Our initial observation of CS/DS as a key factor in the provisional matrix during wound healing by the quantification and characterization of GAGs before and after AB was supported by evidence of elevated levels of certain CSPGs. Although existing research documents the expression of CSPG versican in embryonic chick and rat corneas, as well as during postnatal corneal development, its expression after a corneal injury is not as well documented [62,63]. At present, at least four splice variants of versican, VO, V1, V2, and V3 have been identified [63,64,65,66,67]. These variants are generated as a result of alternative RNA splicing in the two exons encoding the GAG attachment sites, exon 7 which encodes α-GAG and exon 8 which encodes β-GAG [63,65,66,68]. VO, the longest variant, has both exons, while V1 has βGAG, V2 has αGAG, and V3 has neither (Figure 5K) [63,66,68]. Koga et al. demonstrated the expression pattern of VO-V3 in rat corneas, where they illustrated a rapid decrease of all four isoforms at the mRNA and protein levels during postnatal development [63]. Thus, all four versican variants were highly expressed in the cornea at birth but became undetectable in adults except in the limbus [63,69]. Our data corroborate these findings; in the uninjured mouse cornea versican is expressed primarily in the limbal region, with very low versican in the central and peripheral cornea. Interestingly, we found a significant increase in versican expression throughout the cornea and limbus after AB. Functionally, versican has been shown to be an important component during inflammatory responses, such as in infections, certain deceases, and after injuries [70,71,72,73]. Its importance appears to be in its ability to interact with inflammatory cells and other inflammatory components and regulate their availability and activity [74]. We observed all four versican isoforms in adult mouse corneas. Our findings demonstrate an increase in the levels of VO, VI, and V3, and not V2 during corneal wound healing. Although the presence of versican variants has been reported in the mouse retina, vitreous humor and trabecular meshwork [75,76], to our knowledge, this is the first documentation of versican in the adult mouse cornea pre and post injury.

Biglycan, a CS/DSPG known to play an important role during inflammatory responses, was also elevated in the provisional matrix after corneal AB [77,78]. Upon injury or during inflammatory processes, biglycan serves as a ligand for the innate immunity receptors, Toll-like receptors 2 and 4 (TLR2 and TLR4) that are found on macrophages [77,78]. This, in turn, triggers the activation of signaling pathways via p38, p42/44 and NF-κB, with subsequent generation of TNF-α and macrophage inflammatory protein-2 (MIP-2) [78]. Thus, the increase in biglycan expression 5 and 14 days after AB indicates it could play a role in the corneal inflammatory response to chemical injuries. A hallmark of microbial-induced cornea inflammation is activation of TLRs, which then triggers neutrophil infiltration in an attempt to clear invading organisms and prevent dissemination [79,80,81,82]. Thus, these receptors are necessary for ocular surface immune response to infectious agents. We hypothesize TLRs could also be triggered by biglycan to mount an immune response in the context of a corneal chemical injury, such as an AB. In contrast, decorin expression was decreased 5 and 14 days after AB. Decorin and biglycan have both been shown to have a similar spatial distribution in the cornea; however, with distinct temporal expression patterns [83]. Decorin and biglycan both play important roles in regulating collagen fibrillogenesis, with reportedly overlapping functions [83]. Specifically, compound decorin/biglycan-*null* mice present more severe disruption in fibril structure and organization when compared to single *null* mice [83]. At low concentrations, decorin has been shown to be a more efficient regulator of collagen fibrillogenesis when compared to biglycan [83]. Previous studies have suggested that corneal scarring and certain chronic pathologic conditions lead to an up-regulation of both decorin and biglycan bearing highly sulfated DS [84]. To the best of our knowledge this is the first study to look at the expression of decorin and biglycan following corneal chemical injury.

An increase in aggrecan expression was also noted after AB, primarily in the corneal epithelium, basement membrane, surrounding stromal cells, and in the corneal endothelium. Previous studies have identified aggrecan as an important constituent in the scarring process of the fibrotic heart, liver, brain, spinal cord, and skin [85,86,87]. Curiously, aggrecan has previously been shown to associate with lumican in the aging human sclera [88]. Though aggrecan is present in normal human and murine skin, it has been observed to accumulate in scar tissues [89,90,91]. This aggrecan accumulation appears to hinder cell migration, and, thus, prevents proper wound healing, which results in increased scarring [90]. Thus, proper aggrecan turnover is required for effective post wound tissue restoration. To this end, aggrecan has been considered a potential target for reducing scar formation after skin injuries, and, based on our data, this could also be considered for corneal injuries.

The importance of small leucine-rich proteoglycans (SLRPs) in maintaining corneal transparency has been accepted for decades with a vast number of publications on this topic since early 1970s [92,93,94,95,96]. As previously mentioned, normal corneal ECM is composed primarily of KSPGs, and worthy of note are three prominent SLRP members; Lumican, keratocan, and mimecan [16,97]. Studies have revealed that in the healthy cornea, KSPGs are expressed primarily by stromal keratocytes [95,98,99,100]. Given that keratocytes are the main cell type responsible for KSPG synthesis, the trans-differentiation of keratocytes post injury could explain the observed decrease in KSPG expression during the wound healing process. Studies have revealed that the KS content within the corneal stroma is at least one order of magnitude more abundant than in any other tissue [92,100,101]. Conversely the other GAG subtypes, CS/DS and HS, appear to be less abundant [93]. The composition of GAGs and PGs within the corneal stromal ECM is essential for maintaining cornea transparency [100]. KS has been shown to be highly expressed in the murine cornea and limbal region, although it is more prevalent in the former [102]. This study also demonstrated a differential distribution of highly and lesser sulfated KS in the different corneal compartments, with primarily highly sulfated KS being expressed in the epithelium and basement membrane, while both the highly and lesser sulfated forms were expressed in the Bowman’s layer, stroma and Descemet’s membrane [102]. Our study corroborates these findings, demonstrating that both the highly and lesser sulfated forms of KS are expressed primarily in the central cornea when compared to the limbal region. In the central cornea, both forms are highly expressed in the stroma and Descemet’s membrane. Upon corneal injury, keratocytes are rapidly activated, leading to their trans-differentiation into fibroblasts and myoblasts-like cells, which is central to the wound healing process [35,36,97]. Research has previously demonstrated a decrease in KS, and, an overall change in GAG composition after corneal injury [93,94,103]. Cintron et al. analyzed the chemical properties of PGs expressed in rabbit corneal scars from a central 2-mm-diameter full-thickness button excision. They observed the appearance of an unusually large DSPG, an overall decrease in sulfated PGs, and a change in matrix from collagen fibrils to one that is fibrin in nature [60]. In contrast, we found that overall, KS is upregulated in all corneal compartments after AB, although solely the highly sulfated form is upregulated in the corneal epithelium. Moreover, our study revealed a significant increase in sulfated CS and a decrease in lesser-sulfated CS after AB in mice. Another study analyzed the expression of GAGs in rabbit corneas after corneal button excisions over time and found that scars as old as 1 year present elevated levels of CS, and the higher levels of CS are correlated with corneal opacification and loss of transparency [93]. However, corneal opacification subsided when the KS level was restored [93]. Interestingly, macular corneal dystrophy (MCD), a noninflammatory clouding of the cornea, has been linked to mutations in the carbohydrate sulfotransferase 6 gene (CHST6) involved in the biosynthesis of sulfated KS [28,104,105,106]. The loss of this gene results in reduced levels of sulfated KS within the cornea and this loss of sulfated KS, leads to poorly structured collagen fibrils and, consequently, corneal opacification [28,98,107,108,109,110,111]. Thus, the decrease in KS after corneal wounding can be associated with the loss of corneal transparency.

Curiously, in our study no changes in the structural composition of HS chains following AB were noted. We therefore did not investigate the expression profile of HSPGs further. However, our study did find that in the cornea HS is generally expressed in a lesser sulfated form, with an average of one sulfate per disaccharide unit. In the cornea, HS serves to maintain tight junctions of the corneal epithelium, which is crucial to maintaining its structural integrity [112]. Studies using both mice and other model organisms have documented the importance of post-synthesis modification of HS chains by extracellular endosulfatases, SULF1, and SULF2 in epithelial cell migration during wound repair [113,114,115]. The SULFs remove sulfate from the C-6 internal glucosamine residues of intact HSPGs, which liberates HS-associated ligands such as cytokines and growth factors and allows for them to participate in wound repair [114,115,116]. A prior study by our group utilizing a corneal epithelium specific conditional knockout mouse revealed the importance of HS in both corneal homeostasis and wound healing [112]. The same study observed no significant corneal defects when HS in the stroma was targeted. Other studies, such as those performed by the Stepp and Inomata groups, targeting specific HSPGs, syndecan and perlecan, respectively, also show HSPGs expressed in the corneal epithelium are essential for corneal homeostasis and wound healing [117,118,119,120].

Taken together, our study reveals significant changes in the composition of the ECM following a corneal chemical injury. Significant changes in the expression of CSPGs/DSPGs and KSPGs were observed after AB, whereas no significant changes in HS were noted. Importantly, the composition of the provisional matrix changes throughout the 2-week period following AB, therefore, scar tissue is actively being deposition and remodeled during this same time-frame. Thus, understanding the dynamics of the scarring process is vital for planning interventions for preventing corneal scarring.

## 4. Materials and Methods

### 4.1. Animals

Forty 7-week-old C57BL/6J male mice were purchased from Jackson Laboratory (Stock number 000664). Upon arrival, mice were housed in a temperature-controlled facility with an automatic 12-h light–dark cycle at the Animal Facility of the University of Houston. The mice were allowed a week to acclimate to their new environment. All animal related experimental procedures, handling and surgeries were previously approved by the Institutional Animal Care and Use Committee (IACUC) at the University of Houston under protocol 16-044. Animal care and use conformed to the ARVO Statement for the Use of Animals in Ophthalmic and Vision Research.

### 4.2. Alkali Burn Model

Alkali burn (AB) injuries were performed on mice as previously described [121,122]. Briefly, mice were anesthetized with a combination of ketamine (80 mg/kg; Vedco INC, Catalog# 07-890-8598) and xylazine (10 mg/kg; Akorn INC, Catalog# 07-808-1947). Circular 3MM chromatography paper (Whatman, 1-mm diameter; Sigma-Aldrich Corp., St. Louis, MO, USA) was immersed in freshly prepared 0.1 M sodium hydroxide solution (NaOH) and placed on the central cornea of the right eye of an anesthetized mouse for 1 min and 20 s, after which the eye was exhaustively washed with sterile PBS in a dropwise manner for 1 min. Thereafter, excess PBS was dried and a drop of antibiotic ointment Terramycin (Zoetis, NADA # 8-763) was placed onto the injured eye and mice placed on a heating pad for monitoring until they regained consciousness. Mice were culled either 5 or 14 days after injury and corneas collected and either stored at −80 °C for mRNA or GAG extraction or fixed in 4% paraformaldehyde for histological analysis. The left uninjured eye was used as a control for each time point.

### 4.3. In Vivo Confocal Microscopy

Corneal haze, scarring and the presence of inflammatory cells within the cornea were analyzed by in vivo confocal microscopy, as described previously, using a Heidelberg Retinal Tomograph-HRTII Rostock Cornea Module (HRT-II, Heidelberg Engineering Inc., Heidelberg, Germany) [42,123]. For such, mice were anesthetized with a combination of ketamine and xylazine, as described above, and placed in a mouse holder that was designed by us at the college of Optometry, University of Houston. GenTeal gel (Novartis Pharmaceuticals Corp.) was applied to both the eyeball and the tip of the HRT-II objective as an immersion fluid. Subsequently, a series of 40 images were collected starting at the outer epithelial layer through the whole stromal thickness and ending at the corneal endothelium as a continuous z axis scan at 2-μm increments. Corneas of mice 5 and 14 days after AB had an increase in corneal thickness as a consequence of edema, and, therefore, required the collection of two sequential series through the cornea in order to encompass the entire thickness. The lens of the HRT-II has a working distance of 77 μm. Images were exported as a sequence of tiff files and analyzed using ImageJ (Fiji Is Just Image J, an open-source platform for biological-image analysis). Five mice were analyzed per experimental point and a representative profile presented in the figure.

### 4.4. RNA Extraction and Real-Time PCR Analysis

Injured corneas (AB) or uninjured corneas (Ctr) corneas were isolated from five mice per time point, pooled and snap frozen on dry ice and stored in a −80 °C freezer. Total RNA was isolated from these tissue samples using Trizol^®^ Reagent (Invitrogen, Carlsbad, CA, USA) followed by chloroform extraction (Sigma-Aldrich, Catalog#650498). RNA concentration and purity were determined using a spectrophotometer at 260 and 280 nm. First strand cDNA was reverse transcribed using 1.5 to 2 μg of total RNA with the high capacity cDNA Reverse Transcription kit (Applied Biosystems, catalog# 4368814, lot 00593854, Foster City, CA, USA) according to the manufacturer’s instructions. Quantitative real-time PCR (qPCR) amplification was performed on 1 μL or 50 ng of cDNA (1:5) using the PowerUp SYBR Green Master Mix kit (Applied Biosystems, Catalog# A25918) using a CXF Connect Real-time System from BIO-RAD, with an activation cycle of 95 °C for 10 min, 40 cycles of 95 °C for 15 s and 60 °C for 1 min. A complete list of primers used in this study is shown in Table 4. The specificity of amplified products was analyzed through dissociation curves generated by the equipment yielding single peaks. Gene expression levels were normalized against both *Actb* and *Gapdh* using both 2^−ΔCt^ and 2^−ΔΔCt^ methods. The expression profile of AB-injured corneas was analyzed against the expression profile of uninjured control corneas for each time point.

### 4.5. Glycosaminoglycan Extraction from Corneas

AB or Ctr corneas were obtained from seven mice per time point, pooled, snap frozen on dry ice and stored in the −80 °C freezer. For GAG extraction, corneas were homogenized using a repeated combination of low-intensity sonication and pronase/benzonase digestion. After the corneas were completely homogenized, lipids were removed with acetone. The samples were then suspended in 2 mL 0.1 M Tris-HCl, pH 8.0, containing 2 mM CaCl_2_ and 1% Triton X-100 and briefly sonicated (Branson). Thereafter, pronase was added to a final concentration of 0.8 mg/mL, and samples incubated at 50 °C in a shaker for 24 h. Samples were sonicated again, an additional 1.6 mg of pronase added and digestion continued for a further 24 h. Finally, the enzyme was inactivated at 100 °C for 15 min. The buffer was adjusted to 2 mM MgCl_2_, and benzonase (100 mU) was added for 2 h at 37 °C. The enzyme was inactivated at 100 °C for 15 min and undigested material removed by centrifugation for 1 h at 4000× *g*. The supernatant was applied to a DEAE-Sepharose-micro column, washed with ~10 column volumes of loading buffer (~pH 8 Tris Buffer, 0.1 M NaCl) and a GAG fraction eluted in 2 M ammonium acetate. The acetate salt was removed via lyophilization and GAGs reconstituted in DI water for analysis.

### 4.6. Characterization of GAGs by Analysis of Disaccharide Composition

The structural composition of CS/DS and HS/HEP was done via disaccharide characterization with strong anion chromatography (SAX) using high-pressure liquid chromatography (HPLC) at the Complex Carbohydrate Research Center (CCRC) at the University of Georgia (Appendix A). Given the limited amount of material obtained from the mouse corneas, samples were subjected to a single instrumental run, as previously shown [124,125]. Briefly, SAX-HPLC analysis was carried out using a 4.6 × 250 mm analytical column (Waters Spherisorb) with 5 μm particle size at 25 °C using an Agilent system. Disaccharides were eluted with a gradient from 97% of 2.5 mM sodium phosphate, pH 3.5 and 3% of 2.5 mM sodium phosphate, 1.2 M NaCl, pH 3.5 to 100% of 2.5 mM sodium phosphate, 1.2 M NaCl, pH 3.5 over a 55-min period at 1 mL/min. Disaccharides were detected by post-column derivatization by combining the eluent at a 1:1 ratio with 0.25 M NaOH and 1% 2-cyanoacetamide pumped at a flow rate of 0.5 mL/min from a binary HPLC pump. The mixture was heated to 120 °C in a 10-m reaction coil, cooled in a 50-cm cooling coil and directed into a Shimadzu fluorescence detector (λex = 346 nm, λem = 410). Elution profiles were compared to that of commercial standard disaccharides (Dextra Laboratories).

### 4.7. Immunohistochemistry

Eyeballs were enucleated, fixed via immersion in 4% buffered paraformaldehyde, and processed for paraffin embedding (3 corneas) or cryosectioning (3 corneas), as previously described [126,127]. Four-micrometer sections were obtained using a Leica RM 2235 microtome (Leica, Buffalo Grove, IL, USA) or 10 μm sections were obtained using a Leica CM 1950 (Leica, Buffalo Grove, IL, USA) cryostat and mounted on superfrost slides (VWR, Catalog#48311-703). Upon use, the paraffin processed slides were heated at 65 °C for 30 min, and, subsequently, sections were washed with citrisolv (Dicon Labs Inc., Cat# 1601, Gainesville, FL, USA) and rehydrated with subsequent washes of decreasing concentrations of ethanol. Alternatively, sections were incubated for 30 min at 60 °C, and excess tissue embedding medium removed with PBS. All sections were then treated with 0.1% glycine (Fisher Chemical, Catalog#G46-500) for one minute, and nonspecific protein binding sites blocked with 5% FBS (Seradigm, Catalog#3100-500) prepared in PBS. Sections were incubated with anti-CS (clones 1-B-5, 3B3 and 2-B-6), anti-versican (AF3054 that recognizes the V0 isoform from R&D systems), anti-versican (ab177480, raised against a synthetic peptide in the G3 domain, from Abcam), anti-aggrecan (AB1031 from Millipore Sigma), anti-biglycan (clone BigN PR8A4), anti-KS (clones 5-D-4 and 1-B-4), anti-lumican (clone 1F12B10), and hyaluronan binding protein (HABP-385911, Millipore). Sections were washed in PBS and incubated with appropriate secondary antibodies conjugated with Alexa Fluor^®^ 488 (Life Technologies) or Alexa Fluor^®^ 555 (Life Technologies, Carlsbad, CA, USA), or NeutrAvidin^®^Alexa 555 (Life Technologies) in the case of HA, for 2 h at room temperature. The tissues were then washed, permeabilized with 0.1M saponin in PBS and f-actin stained with phalloidin conjugated with Alexa Fluor^®^ 647 (A22284—ThermoFisher Scientific, Waltham, MA, USA) and nuclei stained with 4′,6-diamidino-2-phenylindole (DAPI, Sigma-Aldrich, St. Louis, MO, USA). Sections were mounted in Prolong^®^Gold (Molecular Probes, Eugene, OR, USA) and imaged using a ZEISS LSM 800 Confocal microscope with Airyscan. Images were analyzed using Zen Software (Zeiss). Secondary controls were carried out with a goat IgG isotype control (ab37388; Abcam) in lieu of the primary antibody and did not yield any significant staining (results not shown).

### 4.8. Statistical Analysis

All experiments were carried out with at least five mice per experimental point and values are presented as means ± standard error of the mean. Image analysis and quantification were performed masked to avoid bias. Differences were assessed by t-test or ANOVA, followed by post hoc test for multiple comparisons considering *p* ≤ 0.05 as statistically significant. Statistical analysis was performed with the GraphPad Prism version 5 software package (GraphPad Software, San Diego, CA, USA).

## Figures and Tables

**Figure 1 ijms-22-05708-f001:**
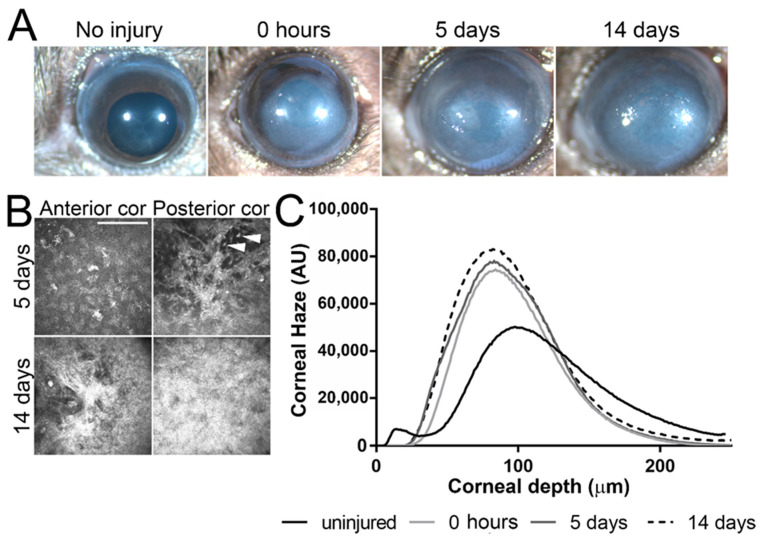
Corneal morphology and haze after AB. Corneas were imaged under a stereomicroscope prior to and immediately, 5 days and 14 days after AB, and, a representative image shown (**A**). Corneas were analyzed by in vivo confocal microscopy at 0, 5, and 14 days after AB and a representative images of the anterior and posterior stroma shown (**B**). Scale bar 50 μm. Inflammatory cells are present at 5 days after AB, indicated with a white arrowhead. In order to evaluate corneal transparency, a z-stack of images was acquired through the entire corneal stroma at 2 μm increments and corneal haze quantified in each frame using Fiji (Image J-win64 by NIH) and plotted as a histogram. Representative histograms are presented (**C**). All mice included in this study were analyzed under a stereomicroscope at each indicated time point and five mice from each experimental group were analyzed by in vivo confocal microscopy at each indicated time point.

**Figure 2 ijms-22-05708-f002:**
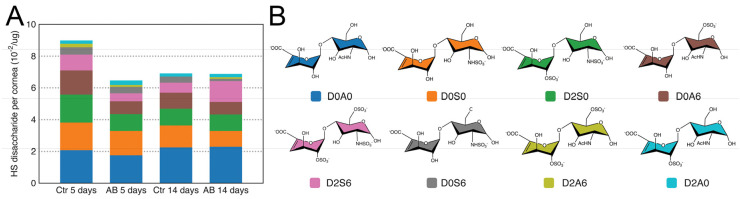
Disaccharide composition of HS in mouse corneas after AB. Total GAGs were extracted from seven pooled corneal tissues and digested with Heparinase in order to generate HS disaccharides. These were then separated by strong anion exchange chromatography using a high pressure liquid chromatography system (SAX-HPLC). The eluent was subjected to fluorescent post-column derivatization using 2-cyanoacetamide in the presence of NaOH. The disaccharide separation profiles were compared to the separation of standard disaccharides for identification, and the abundance of the different HS disaccharides in the injured and uninjured corneas represented (**A**). Representation of the chemical structure of the HS disaccharides identified in the corneal samples before and after AB (**B**).

**Figure 3 ijms-22-05708-f003:**
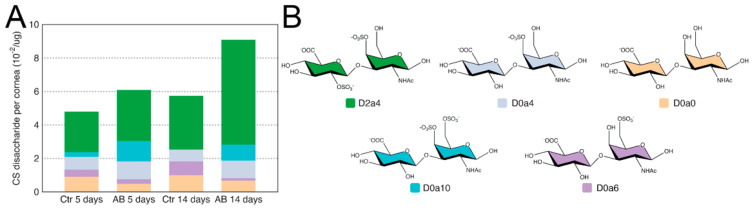
Disaccharide composition of CS/DS in mouse corneas after AB. Total GAGs were extracted from seven pooled corneal tissues and digested with Chondroitinase ABC in order to generate CS and DS disaccharides. These were then separated by strong anion exchange chromatography using a high pressure liquid chromatography system (SAX-HPLC). The eluent was subjected to fluorescent post-column derivatization using 2-cyanoacetamide in the presence of NaOH. The disaccharide separation profiles were compared to the separation of standard disaccharides for identification and quantification, and, the abundance of the different CS/DS disaccharides in the injured and uninjured corneas represented (**A**). Representation of the chemical structure of the CS/DS disaccharides identified in the corneal samples before and after AB (**B**).

**Figure 4 ijms-22-05708-f004:**
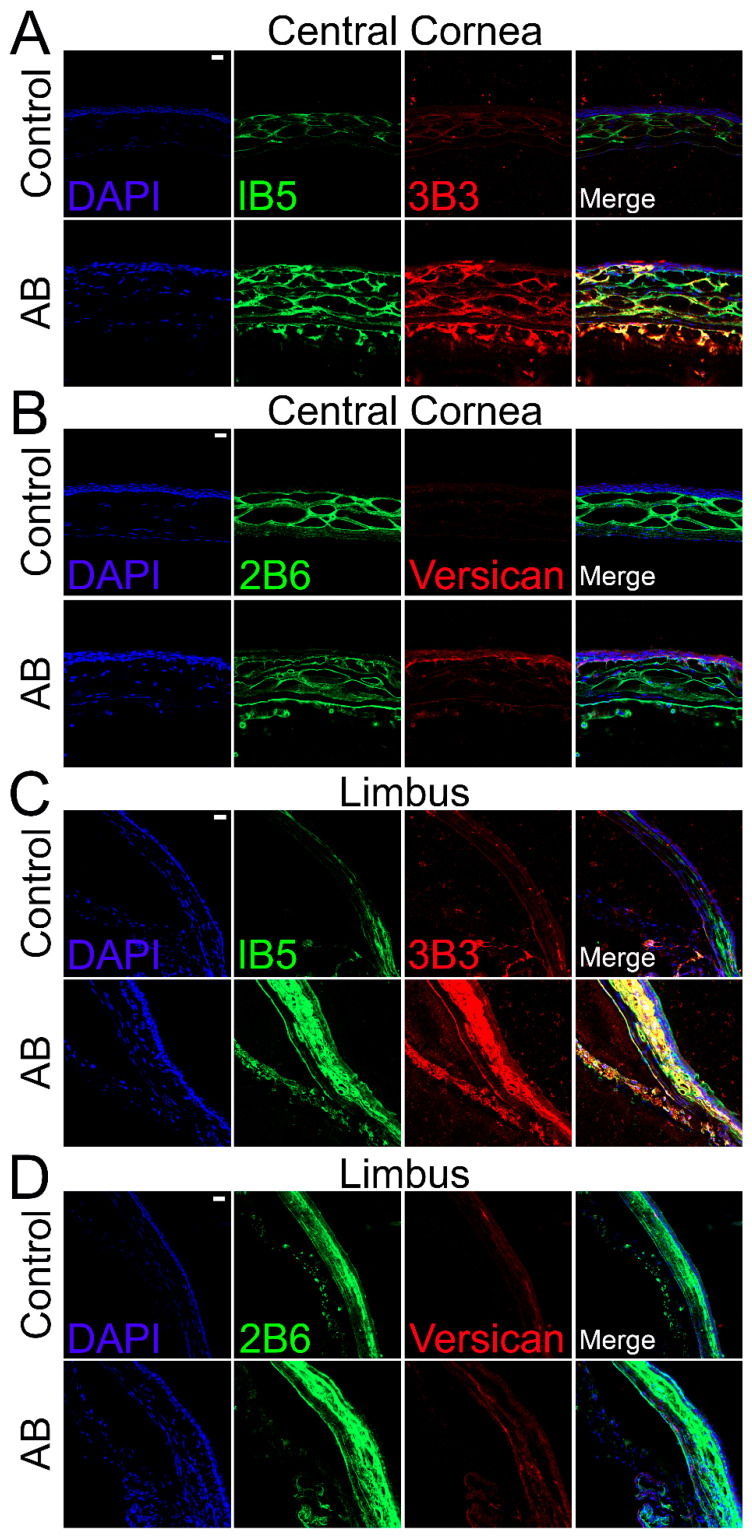
Expression and localization of different CS epitopes and versican in the central cornea and limbal region of mice before and after AB. Mouse corneas were subjected to AB and corneas analyzed after 14 days, with uninjured contralateral corneas used as controls. The central cornea (**A**,**B**) and the limbal region (**C**,**D**) were imaged under a confocal microscope. The expression and localization of CS epitopes were analyzed using 1B5 and 3B3 antibodies by immunocytochemistry (**A**,**C**). The expression and localization of the CS epitope 2B6 and versican were analyzed by immunocytochemistry (**B**,**D**). Nuclei were stained with DAPI (blue). Scale bar 20 μm. Seven corneas were analyzed per experimental group and a representative image displayed.

**Figure 5 ijms-22-05708-f005:**
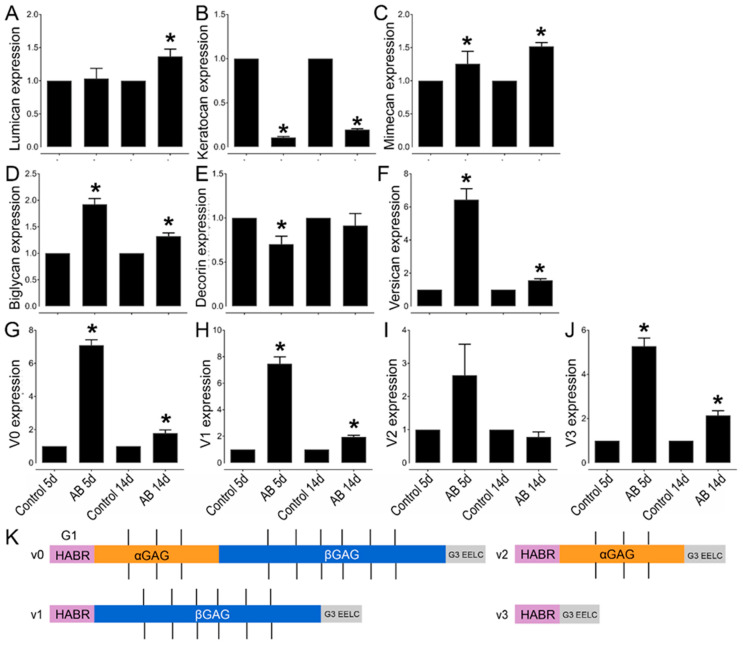
Relative expression profile of CSPGs and KSPGs in mouse corneas after AB. Mouse corneas were subjected to AB and analyzed after 5 and 14 days, compared to the contralateral uninjured eye. Real-time PCR (qPCR) analysis was done to verify the expression levels of lumican (**A**), keratocan (**B**), mimecan (**C**), biglycan (**D**), decorin (**E**), versican (**F**), V0 (**G**), V1 (**H**), V2 (**I**), and V3 (**J**). The relative expression levels were normalized against the arithmetic mean of GAPDH and beta-actin using the *2*^−ΔΔCt^ method. * represents *p* ≤ 0.05. Five corneas were analyzed per experimental group. Schematic representation of the different domains of versican spliced variants (**K**). The different variants bind to HA via the G1 domain (HABR—pink). The CS-bearing domains GAGα (orange) and GAGβ (blue) are represented with the CS chains as black straight lines. Finally, the G3 domain (gray) contains epidermal growth factor (EGF)-like repeats, a lectin-like motif and another HA binding domain, enabling further interactions with HA [44,45]).

**Figure 6 ijms-22-05708-f006:**
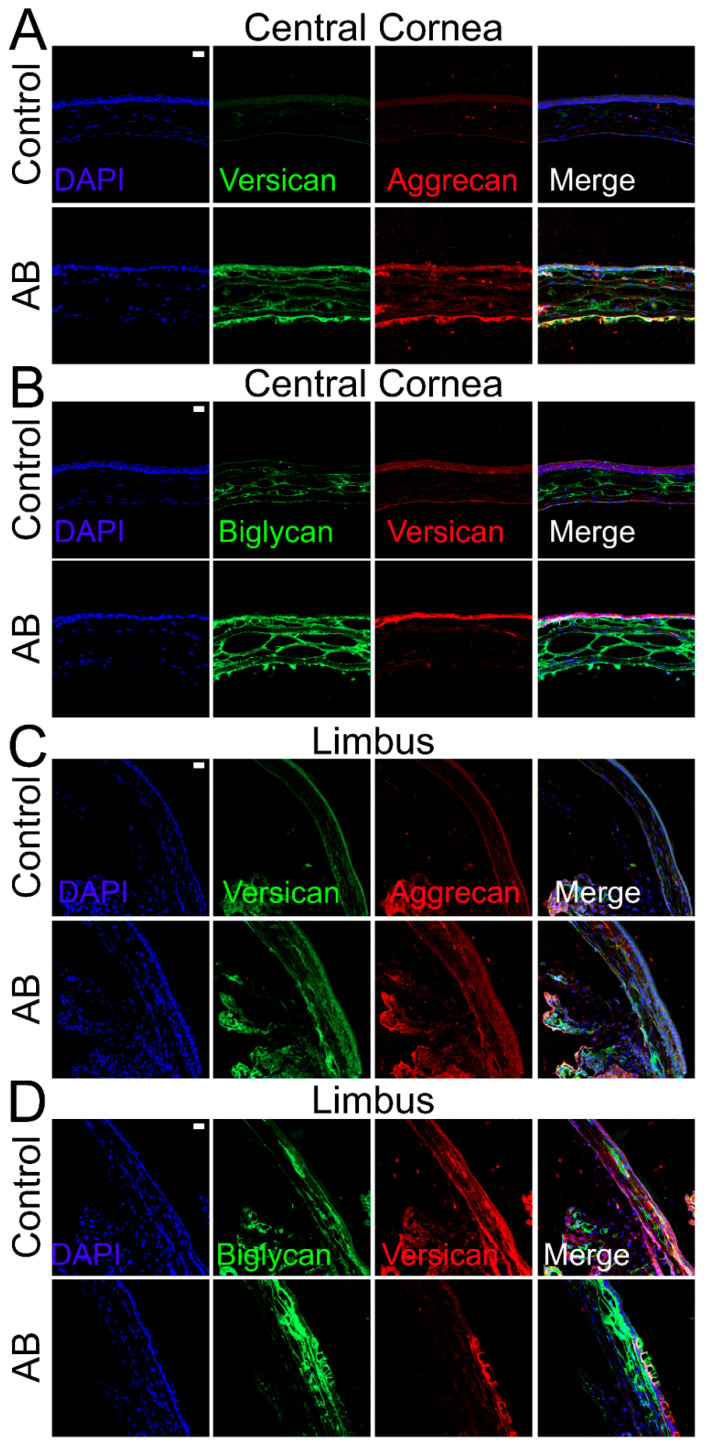
Expression and localization of versican, aggrecan and biglycan in the central corneal and limbal region of mice before and after AB. Mouse corneas were subjected to AB and corneas analyzed after 14 days, with contralateral corneas used as controls. The central cornea (**A**,**B**) and the limbal region (**C**,**D**) were imaged under a confocal microscope. The expression and localization of versican and aggrecan (**A**,**C**), and, biglycan and versican (**B**,**D**) were analyzed using immunocytochemistry. Nuclei were stained with DAPI (blue). Scale bar 20 μm. Seven corneas were analyzed per experimental group and a representative image displayed.

**Figure 7 ijms-22-05708-f007:**
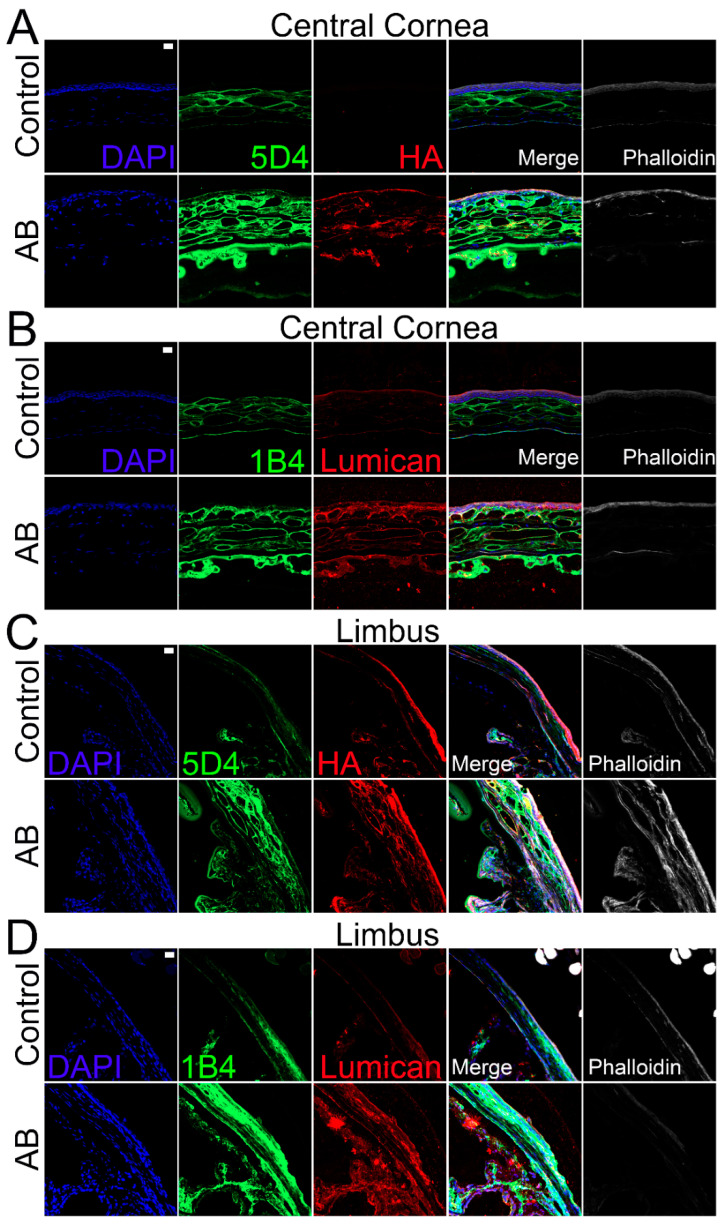
Expression and localization of different KS epitopes, HA and lumican in the central corneal and limbal region of mice before and after AB. Mouse corneas were subjected to AB and corneas analyzed after 14 days, with contralateral corneas used as controls. The central cornea (**A**,**B**) and the limbal region (**C**,**D**) were imaged under a confocal microscope. The expression and localization of highly sulfated KS (5D4) and HA were analyzed using immunocytochemistry (**A**,**C**). The expression and localization of lesser sulfated KS (1B4) and lumican were analyzed using immunocytochemistry (**B**,**D**). Nuclei were stained with DAPI (blue). Scale bar 20 μm. Seven corneas were analyzed per experimental point and a representative image displayed.

**Figure 8 ijms-22-05708-f008:**
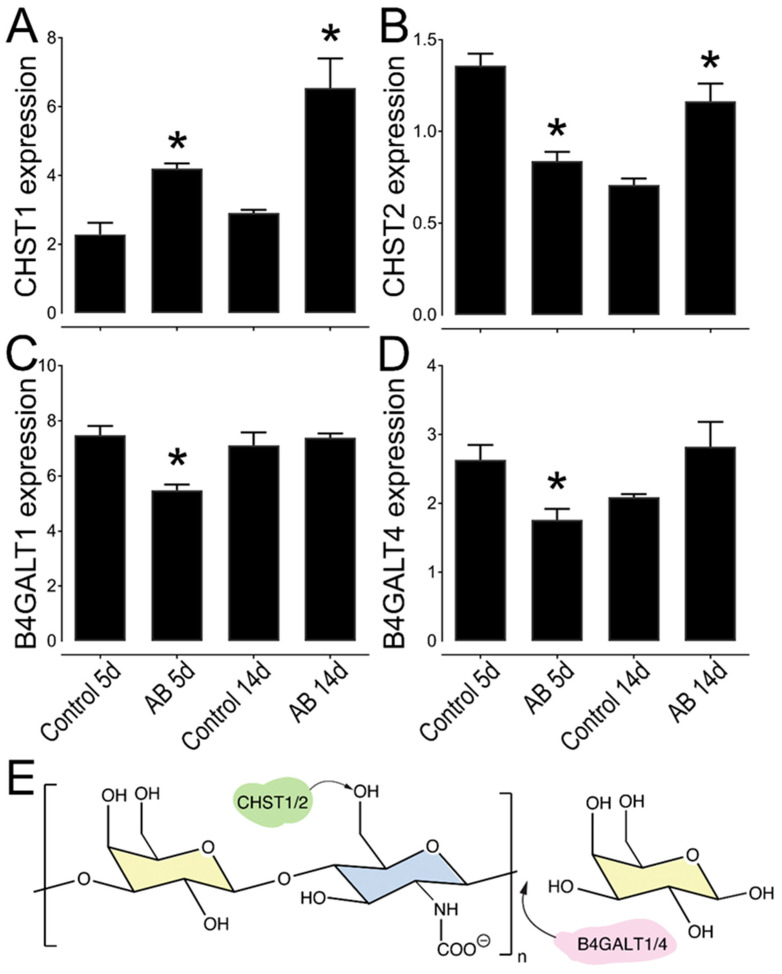
Relative expression profile of KS biosynthetic enzymes in mouse corneas after AB. Mouse corneas were subjected to AB and analyzed after 5 and 14 days, compared to the contralateral uninjured eye. Real-time PCR analysis (qPCR) was done to verify the expression levels of CHST1 (**A**), CHST2 (**B**), B4GALT1 (**C**), and B4GALT4 (**D**). The relative expression levels were normalized against GAPDH using the *2*^−ΔCt^ method. * represents *p* ≤ 0.05. Five corneas were analyzed per experimental group. Schematic representation of the activity of each enzyme analyzed, CHST1 and 2 catalyze the transfer of a sulfate to the position 6 of galactose, while B4GALT1 and 4 catalyze the transfer of a galatose residue to the polymerizing KS chain (**E**).

**Table 1 ijms-22-05708-t001:** Heparin/heparan sulfate content in corneas of wild-type mice 5 and 14 days after AB.

	Ctr 5 Days		AB 5 Days		Ctr 14 Days		AB 14 Days	
HS								
D0A0	2.081633	*23*	1.755102	*27*	2.244898	*32*	2.285714	*33*
D0S0	1.734694	*19*	1.530612	*24*	1.387755	*20*	1	*15*
D2S0	1.755102	*20*	1.061224	*16*	1.061224	*15*	1.040816	*15*
D0A6	1.530612	*17*	0.816327	*13*	1	*14*	0.795918	*12*
D2S6	1	*11*	0.489796	*8*	0.632653	*9*	1.306122	*19*
D0S6	0.44898	*5*	0.408163	*6*	0.387755	*6*	0.142857	*2*
D2A6	0.22449	*3*	0.122449	*2*	*ND*	*0*	0.102041	*1*
D2A0	0.204286	*2*	0.282857	*4*	0.195714	*3*	0.205714	*3*
Total HS	8.979796	*100*	6.466531	*100*	6.91	*100*	6.879184	*100*

Values represent μg per cornea *Percentage* (*w*/*w*). ND = below the limits of detection.

**Table 2 ijms-22-05708-t002:** Chondroitin/dermatan sulfate content in corneas of wild-type mice 5 and 14 days after AB.

	Ctr 5 Days		AB 5 Days		Ctr 14 Days		AB 14 Days	
**CS**								
D0a0	0.914285714	*19*	0.485714286	*8*	1	*17*	0.671428571	*7*
D0a6	0.414285714	*9*	0.271428571	*4*	0.814285714	*14*	0.142857143	*2*
D0a4	0.771428571	*16*	1.071428571	*18*	0.714285714	*12*	1.057142857	*12*
D0a10	0.271428571	*5*	1.214285714	*20*	*ND*		0.957142857	*11*
D2a4	2.428571429	*51*	3.042857143	*50*	3.214285714	*56*	6.257142857	*69*
Total CS	4.8	*100*	6.085714286	*100*	5.757142857	*100*	9.1	*100*

Values represent μg per cornea. *Percentage* (*w*/*w*). ND = below the limits of detection.

**Table 3 ijms-22-05708-t003:** Identification of KS biosynthetic enzymes by RNAseq analysis.

Gene	Name	Mean Count	Log 2 Fold Change	Log Fold Change	*p*-Value
Fut8	glycoprotein 6-alpha-L-fucosyltransferase	303.5	−0.374	0.610	0.540
B3gnt2	N-acetyllactosaminide beta-1,3-N-acetylglucosaminyltransferase	592.6	0.133	0.555	0.811
St3gal1	beta-galactoside alpha-2,3-sialyltransferase	734.4	−1.302	0.686	0.0578
St3gal3	neolactotetraosylceramide alpha-2,3-sialyltransferase	143.1	−0.890	0.699	0.203
Chst1	keratan sulfate 6-sulfotransferase 1	26.8	−0.818	0.608	0.178
St3gal2	beta-galactoside alpha-2,3-sialyltransferase	792.2	0.0791	0.541	0.884
Chst2	carbohydrate 6-sulfotransferase 2	71.4	−0.299	0.706	0.672
B4galt1	beta-1,4-galactosyltransferase 1	2806.6	−0.0580	0.509	0.909
B4galt2	beta-1,4-galactosyltransferase 2	110.6	−0.836	0.707	0.237
B4galt3	beta-1,4-galactosyltransferase 3	866.6	−0.796	0.616	0.196
B4galt4	beta-1,4-galactosyltransferase 4	59.6	−0.536	0.716	0.455
B4galt7	beta-1,3-N-acetylglucosaminyltransferase 7	205.2	−0.244	0.629	0.698
B3gnt7	solute carrier family 35 (UDP-N-acetylglucosamine (UDP-GlcNAc) transporter)	291.7	−0.224	0.601	0.657
Slc35a3	glycoprotein 6-alpha-L-fucosyltransferase	1062.6	0.743	0.603	0.218

**Table 4 ijms-22-05708-t004:** Primers used for real time PCR analysis.

Gene (Mus Musculus)	Accession Number	Forward (5⟶3′)	Reverse (5′⟶3′)	Product Length
*Mimecan*	NM_008760.5	TTTGCAGACATGCCAAACCT	AGCTTTGGAGGAAGAACTGGA	152
*Keratocan*	NM_008438.3	TAGCTAACCTAACACCAGCCA	GGTTGCCATTACAGCACCTTG	70
*Lumican*	*NM_008524.2*	*CCCACCCTGACAGAGTTCAC*	*CAGCAAGTCCTCTGTGACCTTA*	112
*Byglican*	*NM_007542.5*	*TGTCCCTCCCCAGGAACATT*	*GTCCCAGAAACCCTTCTGCT*	102
*Decorin*	*NM_001190451.*	*AACTGTGCTATGGAGTAGAAGCA*	*ATCTCATGTATTTTCACGACCTTTT*	192
*Versican*	*NM_001081249.1*	*ACCTACCTTACCACCCAATTAC*	*GTAGTGAAACACAACACCATCCA*	316
*Vcan-Transcript variant 1*	*NM_001081249.1*	*CCAAGTTCCACCCTGACATAAA*	*CACTGCAAGGTTCCTCTTCTT*	129
*Vcan-Transcript variant 2*	*NM_019389.2*	*TGAGAACCAGACATGCTTCC*	*TGAATCTATTGGATGACCACTTACA*	103
*Vcan-Transcript variant 3*	NM_001134474.1	*GGTGAGAACCCTGTATCGTTT*	*GGTGGTTGCCTCTGATATATTCT*	109
*Vcan-Transcript variant 4*	NM_001134475.1	*CAGATTTGATGCCTACTGCTTTAAAC*	*GATAACAGGTGCCTCCGTTGA*	77
*carbohydrate sulfotransferase 1*	NM_018763.2	*CCCCTAGCAGAAGAGAACCG*	*GCTCCGAGAAGGACCTGGAG*	116
*carbohydrate sulfotransferase 2*	NM_001356552.1	*CCTCCCTTCAGGAGCTTCAAA*	*CACACAGCAGTTACCTTCCC*	126
*4-galactosyltransferase, polypeptide 4*	NM_001285793.1	*GGGCTGTGAGCCGGTGAT*	*CGGGGATCTGATGGCAACTC*	80
*4-galactosyltransferase, polypeptide 1*	NM_022305.6	*GGTGGCCATCATCATCCCAT*	*GGTGTCTCCAGCCTGATTGA*	130
*Actb*	NM_007393.5	CACTGTCGAGTCGCGTCC	TCATCCATGGCGAACTGGTG	89
*Gapdh*	NM_001289726.1	AACAGCAACTCCCACTCTTC	CCTGTTGCTGTAGCCGTATT	111

## Data Availability

Data is contained within the article or Appendix A.

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
