# Peer review of "Extracellular Matrix Deposition and Remodeling after Corneal Alkali Burn in Mice"

_ijms, 2021, doi:10.3390/ijms22115708_

Round 1

Reviewer 1 Report

The author revealed significant changes in the composition of the ECM following a corneal chemical injury. This basic research is interesting. However, the method of statistical analysis and experimental number are not clear. Our comments are followings

1) General: Author should mention the experimental number in the Fig. legend. In addition, Author must analyze the statistical difference by Multiple comparison test, such as Dunnett and Tukey, and mention in the Fig. legend.

2) Figure 1C: Author should add the unit in the x- and y-axis. Moreover, please explain how evaluate the curve following corneal. Did you combine or connect some of the numbers calculated by Image J ?. Please attach S.D. or S.E. if necessary.

3) Figure 2A: Author should add the S.D. or S.E.

4) Table 1: Please define the ND. The total number is not 100% (ex. 14d is 99%). Please consider significant digits.

5) Figure 3A: Author should add the S.D. or S.E. Please remove the value between of Fig. and Fig. legend.

6) Table 2: Please add the “0” in the ND. The total number is not 100%. Please consider significant digits.

7) Figure 5: Author should add the S.D. or S.E in the control 5d and 14 d.

8) Table 4: Please move to method (line 570).

9) line 572: “Gene expression levels were normalized against both Actb and Gapdh”. Which of Actb and Gapdh was used as the standard in the Fig. 5 and 8.

10) Please more explain why the CHST2 expression in control 5d was lower than that in control 14 d in the Fig. 8.

11) Please revise “®” to superscript.

Author Response

Reviewer 1

The author revealed significant changes in the composition of the ECM following a corneal chemical injury. This basic research is interesting. However, the method of statistical analysis and experimental number are not clear. Our comments are followings

  • General: Author should mention the experimental number in the Fig. legend. In addition, Author must analyze the statistical difference by Multiple comparison test, such as Dunnett and Tukey, and mention in the Fig. legend.
  1. Originally, the number of mice used in each of the experiments was indicated in the methods sections; however, as per request by the reviewer, we have now also included this information in the figure legends. In reference to the statistical analysis, we apologize for our oversight and have now included a section outlining our statistical analysis at the end of materials and methods.

  • Figure 1C: Author should add the unit in the x- and y-axis. Moreover, please explain how evaluate the curve following corneal. Did you combine or connect some of the numbers calculated by Image J ?. Please attach S.D. or S.E. if necessary.
  1. Again, we apologize for the oversight and have now included the units for the x and y axis in Figure 1C. As now stated in the figure legend, we analyzed five mice per experimental group. Each mouse was analyzed individually, and, thereafter, we chose a representative images and histograms to present in Figure 1. The fact that we presented representative histogram has now been included in the figure legend. We did, however, use the values of all five corneas analyzed per experimental group for statistical analysis. This is similar to what we have previous done with this form of analysis, please refer to Coulson- Thomas et al.,2013 [Stem Cells. 2013 October ; 31(10): . doi:10.1002/stem.1481].

3) Figure 2A: Author should add the S.D. or S.E.

  1. Analysis of CS and HS requires a large amount starting material in reference to the amount of total GAG that can be obtained per cornea, thus, we were required to pool seven corneas per experimental point for analysis. The analysis was carried out at the Complex Carbohydrate Research Center at the University of Georgia; which is the leader in the field for this type of analysis. As mentioned, the data presented in Figures 2 and 3 comprised of whole GAG isolation from seven pooled samples that were subjected to GAG lyase digestion and disaccharide analysis by SAX-HPLC against a set of commercial standards. Due to the low amount of material and detection limits of the analysis only a single instrumental run could be performed on these samples. While multiple replicates are obviously preferable, the trace amounts of GAGs isolated from biological samples, tissues and appropriate pools mean that, frequently, only a single isolation and analysis may be reported in literature as part of an observational study when the total amount of tissue is insufficient, Ticar et al. 2020 and Damron et al. 2021. It is important to note that this form of analysis is extremely reproducible and was carried out in a blinded manner. Thus, we cannot provide SD or SE in Figures 2A and 3A.

    Ticar et al "Biocompatibility and structural characterization of glycosaminoglycans isolated from heads of silver-banded whiting" International Journal of Biological Macromolecules 151 (2020) 663-676

    Damron, CL et al "Offspring of Obese Dams Exhibit Sex-Differences in Pancreatic Heparan Sulfate Glycosaminoglycans and Islet Insulin Secretion" Frontiers in Endocrinology (2021) 12, p 507

4) Table 1: Please define the ND. The total number is not 100% (ex. 14d is 99%). Please consider significant digits.

  1. ND stands for Not detectable (or more correctly, bellow the limit of detection), and this has now been added as a footnote to the tables. We thank the reviewer for pointing out this oversight. In table 1 the values for ctr 5 days, AB 5 days and AB 14 days all equate to 100%, however, indeed the values for ctr 14 days comes to 99%. The percentages were presented without the use decimal points, and, all values were rounded up from 0.5, as is standard practice. The exact values using two decimal points would be 32.48, 20.08, 15.35, 14.47, 9.15, 5.60 and 2.80. As the reviewer can see, we have rounded the values up or down correctly, and, therefore, are unable to change the listed values so that they equate to 100%.

5) Figure 3A: Author should add the S.D. or S.E. Please remove the value between of Fig. and Fig. legend.

  1. We believe that formatting by the editorial office may have introduced the values between Figure 3 and the Figure legend. We believe we have been able to delete them. Thank you for pointing this out. Please see the response above regarding the addition of SD or SE to Figure 3A.

6) Table 2: Please add the “0” in the ND. The total number is not 100%. Please consider significant digits.

  1. We cannot replace the ND for a 0 since the amount was simply beyond the limit of detection for this method, and, therefore, it would not be absolute 0. We have corrected the values and now all equate to 100%.

7) Figure 5: Author should add the S.D. or S.E in the control 5d and 14 d.

  1. For figure 5, the relative change in gene expression for each gene of interest was calculated using the ΔΔCt method, where mean expression of two reference genes (Actb and Gapdh) was used for normalization to obtain the ΔCt values. To determine the fold-change value (2-ΔΔCt), all expression levels for AB values were analyzed in reference to the contralateral control samples which were used as the calibrators and normalized to 1. Essentially, the control for each time point was set as 1 and data from experimental samples were expressed as fold difference compared to control. Therefore, error bars for these controls would just be straight horizontal lines. Please see references https://doi.org/10.1016/j.fsi.2012.12.015 and DOI: 10.1186/1750-1326-8-21. More specifically, the mean expression of the housekeeping genes gapdh and beta actin was used as an internal control to normalize target gene expression between samples using formula ΔCt=Target Ct-Refs mean Ct. The ΔΔCT value of the injured eye was normalized to the control right eye using the formula, ΔΔCT = ΔCT (target gene) – ΔCT (calibrator or control). Which for the control would be ΔΔCT = ΔCT control– ΔCT control, which is equal to zero. The control eye was set as 1 using formula 2^ΔΔCT or 2^(0), which was also used to calculate fold change for experimental eye. For Figure 8, we used the 2-Δct method to analyze the data, and for this type of analysis we calculated the absolute gene expression for control and AB samples independently, and, therefore, all values have an error bar.

8) Table 4: Please move to method (line 570).

  1. This table has been moved to the suggested location in the text.

9) line 572: “Gene expression levels were normalized against both Actb and Gapdh”. Which of Actb and Gapdh was used as the standard in the Fig. 5 and 8.

  1. The expression values of the genes of interest were normalized against mean of both Actb and Gapdh, as mentioned in the manuscript. Please see our response to question 7 regarding calculation summary.

10) Please more explain why the CHST2 expression in control 5d was lower than that in control 14 d in the Fig. 8.

  1. The expression of CHST2 was higher in the contralateral control eye at five days after AB when compared to the contralateral control at 14 days after AB. We are unsure why this was observed for the CHST2 enzyme expression, since all of our other data showed very similar expression levels between the two control samples. However, previous studies have observed certain changes in the contralateral eye after injury which is why we analyzed our two control groups separately.

11) Please revise “®” to superscript.

  1. We have made all of the ®’s superscript.

Reviewer 2 Report

The authors investigated the extracellular matrix deposition in mice after alkaline burn on cornea. This is more like a descriptive study. Results may be useful for future references. Here are my comments:

  1. This is no scale bar in most of the figures including Figure 1B.
  2. Also in Figure 1B, please indicate "inflammatory cell infiltration was clearly evident throughout the corneal stroma".
  3. In Figure 1C, please indicate the sample size and statistics to support the statement of "significant increase in corneal haze".
  4. In Figures 2 and 3, please indicate how many samples were analyzed. Also statistics and error bars are needed.
  5. In the legends of Figures 5 and 8, please make sure the meaning of "*" represents p larger or smaller than 0.05.
  6. For the alkaline burn procedure, please indicate the cornea area receiving alkaline burn was restricted within the limbus or expanded out of limbus. THis is important for the evaluation of results in Figures 4 and 6.
  7. Please explain if the control corneas were collected from the contralateral eyes (left eyes) of mice which the right eyes received alkaline burn.

Author Response

Reviewer 2

The authors investigated the extracellular matrix deposition in mice after alkaline burn on cornea. This is more like a descriptive study. Results may be useful for future references. Here are my comments:

This is no scale bar in most of the figures including Figure 1B.

R. We have added a scale bar to Figure 1B. However, all other figures had a scale bars, albeit they were very small and narrow so hard to see. We have attempted to make them thicker so that they are easier to see in the figures.

Also in Figure 1B, please indicate "inflammatory cell infiltration was clearly evident throughout the corneal stroma".

R. We have added white arrowheads to indicate inflammatory cell infiltration.

 In Figure 1C, please indicate the sample size and statistics to support the statement of "significant increase in corneal haze".

R. We have included this information in the manuscript and in the Figure legend.

In Figures 2 and 3, please indicate how many samples were analyzed. Also statistics and error bars are needed.

R. We have identified how many samples were used in the figure legend and, also, in the manuscript methods section.

In the legends of Figures 5 and 8, please make sure the meaning of "*" represents p larger or smaller than 0.05.

R. We thank the reviewer for pointing out this mistake; this has now been corrected in the figure legends. We have also added a statistical analysis section to the methods section of the manuscript.

For the alkaline burn procedure, please indicate the cornea area receiving alkaline burn was restricted within the limbus or expanded out of limbus. THis is important for the evaluation of results in Figures 4 and 6.

R. We absolutely agree with the reviewer. The AB burn was carried out by placing a 1mm circular filter paper soaked in 0.1M sodium hydroxide onto the center of the cornea. This form of injury was established in lieu of the previously used method of instilling 1 or 2 uL of sodium hydroxide onto the ocular surface, in order to generate a model that spares limbal stem cells. Therefore, this AB injury is intended to be limited to the central and peripheral cornea. This form of injury is widely used in the Winston Kao lab and others labs. The Kao lab (a top lab in the field) has previously thoroughly characterized the severity/extent of the injury. References for this form of injury are included in the manuscript. This form of AB is often referred to as the AB model sparing the limbal region.

Please explain if the control corneas were collected from the contralateral eyes (left eyes) of mice which the right eyes received alkaline burn.

R. Yes, the contralateral control eyes were the left eyes (OS), while the AB samples were right eyes (OD).

Round 2

Reviewer 1 Report

  Most of my concerns are resolved. However, we have some question. Our comments are followings

Q3. Figure 2A: Author should add the S.D. or S.E.

A3. Analysis of CS and HS requires a large amount starting material in reference to the amount of total GAG that can be obtained per cornea, thus, we were required to pool seven corneas per experimental point for analysis. The analysis was carried out at the Complex Carbohydrate Research Center at the University of Georgia; which is the leader in the field for this type of analysis. As mentioned, the data presented in Figures 2 and 3 comprised of whole GAG isolation from seven pooled samples that were subjected to GAG lyase digestion and disaccharide analysis by SAX-HPLC against a set of commercial standards. Due to the low amount of material and detection limits of the analysis only a single instrumental run could be performed on these samples. While multiple replicates are obviously preferable, the trace amounts of GAGs isolated from biological samples, tissues and appropriate pools mean that, frequently, only a single isolation and analysis may be reported in literature as part of an observational study when the total amount of tissue is insufficient, Ticar et al. 2020 and Damron et al. 2021. It is important to note that this form of analysis is extremely reproducible and was carried out in a blinded manner. Thus, we cannot provide SD or SE in Figures 2A and 3A.

     Ticar et al "Biocompatibility and structural characterization of glycosaminoglycans isolated from heads of silver-banded whiting" International Journal of Biological Macromolecules 151 (2020) 663-676

    Damron, CL et al "Offspring of Obese Dams Exhibit Sex-Differences in Pancreatic Heparan Sulfate Glycosaminoglycans and Islet Insulin Secretion" Frontiers in Endocrinology (2021) 12, p 507

Q3-2. Please mention the number in the Fig. legend. Moreover, author should add the above information in the discussion.

Q9. line 572: “Gene expression levels were normalized against both Actb and Gapdh”. Which of Actb and Gapdh was used as the standard in the Fig. 5 and 8.

A9. The expression values of the genes of interest were normalized against mean of both Actb and Gapdh, as mentioned in the manuscript. Please see our response to question 7 regarding calculation summary.

Q9-2 Please show the expression values of the genes as /Actb or /Gapdh in the y-axis.

Q10. Please more explain why the CHST2 expression in control 5d was lower than that in control 14 d in the Fig. 8.

A10. The expression of CHST2 was higher in the contralateral control eye at five days after AB when compared to the contralateral control at 14 days after AB. We are unsure why this was observed for the CHST2 enzyme expression, since all of our other data showed very similar expression levels between the two control samples. However, previous studies have observed certain changes in the contralateral eye after injury which is why we analyzed our two control groups separately.

Q10-2. Author should add the above information in the discussion.

Author Response

Reviewer 1

Most of my concerns are resolved. However, we have some question. Our comments are followings

Q3. Figure 2A: Author should add the S.D. or S.E.

A3. Analysis of CS and HS requires a large amount starting material in reference to the amount of total GAG that can be obtained per cornea, thus, we were required to pool seven corneas per experimental point for analysis. The analysis was carried out at the Complex Carbohydrate Research Center at the University of Georgia; which is the leader in the field for this type of analysis. As mentioned, the data presented in Figures 2 and 3 comprised of whole GAG isolation from seven pooled samples that were subjected to GAG lyase digestion and disaccharide analysis by SAX-HPLC against a set of commercial standards. Due to the low amount of material and detection limits of the analysis only a single instrumental run could be performed on these samples. While multiple replicates are obviously preferable, the trace amounts of GAGs isolated from biological samples, tissues and appropriate pools mean that, frequently, only a single isolation and analysis may be reported in literature as part of an observational study when the total amount of tissue is insufficient, Ticar et al. 2020 and Damron et al. 2021. It is important to note that this form of analysis is extremely reproducible and was carried out in a blinded manner. Thus, we cannot provide SD or SE in Figures 2A and 3A.

Ticar et al "Biocompatibility and structural characterization of glycosaminoglycans isolated from heads of silver-banded whiting" International Journal of Biological Macromolecules 151 (2020) 663-676

Damron, CL et al "Offspring of Obese Dams Exhibit Sex-Differences in Pancreatic Heparan Sulfate Glycosaminoglycans and Islet Insulin Secretion" Frontiers in Endocrinology (2021) 12, p 507

Q3-2. Please mention the number in the Fig. legend. Moreover, author should add the above information in the discussion.

  1. The number of pooled samples was already included in the Figure legend and in the material and methods section. We have also added a statement that samples were subjected to a single instrumental run and included the references listed above in our response in the methods section.

Q9. line 572: “Gene expression levels were normalized against both Actb and Gapdh”. Which of Actb and Gapdh was used as the standard in the Fig. 5 and 8.

A9. The expression values of the genes of interest were normalized against mean of both Actb and Gapdh, as mentioned in the manuscript. Please see our response to question 7 regarding calculation summary.

Q9-2 Please show the expression values of the genes as /Actb or /Gapdh in the y-axis.

  1. We tried to add this information to the y axis of the graphs, but the excess text would mean we would have to significantly decrease the font size, which would jeopardize the ability for individuals to read the text. So instead, we have opted to include this information in the figure legend. Though we appreciated the recommendation, we have listed two references, DOI: 10.1186/s13293-015-0025-y and DOI: 10.1007/s00213-016-4425-4, where similar methods were used and the authors have simply listed “Gene of interest” expression on the y axis.

Q10. Please more explain why the CHST2 expression in control 5d was lower than that in control 14 d in the Fig. 8.

A10. The expression of CHST2 was higher in the contralateral control eye at five days after AB when compared to the contralateral control at 14 days after AB. We are unsure why this was observed for the CHST2 enzyme expression, since all of our other data showed very similar expression levels between the two control samples. However, previous studies have observed certain changes in the contralateral eye after injury which is why we analyzed our two control groups separately.

Q10-2. Author should add the above information in the discussion.

  1. We have added this information to the results section where we discuss the findings of the expression levels of the different KS biosynthetic enzymes.

Reviewer 2 Report

The authors have addressed all my previous comments satisfactorily. I believe the study is now ready to be considered for publication.

Author Response

Reviewer 2

The authors have addressed all my previous comments satisfactorily. I believe the study is now ready to be considered for publication.

R. We thank the reviewer for his time and all the important suggestions/comments made in the first round of revisions.
